# Unified Pretraining Framework for Document Understanding

**Jiuxiang Gu**[1], **Jason Kuen**[1], **Vlad I. Morariu**[1], **Handong Zhao**[1],
**Nikolaos Barmpalios**[2], **Rajiv Jain**[1], **Ani Nenkova**[1], **Tong Sun**[1]
[1]Adobe Research, [2]Adobe Document Cloud
{jigu,kuen,morariu,hazhao,barmpali,rajijain,nenkova,tsun}@adobe.com

## Abstract

Document intelligence automates the extraction of information from documents and supports many business applications. Recent self-supervised learning methods on large-scale unlabeled document datasets have opened up promising directions towards reducing annotation efforts by training models with self-supervised objectives. However, most of the existing document pretraining methods are still language-dominated. We present UDoc, a new unified pretraining framework for document understanding. UDoc is designed to support most document understanding tasks, extending the Transformer to take multimodal embeddings as input. Each input element is composed of words and visual features from a semantic region of the input document image. An important feature of UDoc is that it learns a generic representation by making use of three self-supervised losses, encouraging the representation to model sentences, learn similarities, and align modalities. Extensive empirical analysis demonstrates that the pretraining procedure learns better joint representations and leads to improvements in downstream tasks.

## 1 Introduction

Document intelligence is a broad research area that includes techniques for information extraction and understanding. Unlike plain-text documents in natural language processing (NLP) [1, 2], a physical document can be composed of multiple elements: tables, figures, charts, *etc*. In addition, a document usually includes rich visual information, and can be one of various types of documents (scientific paper, form, resume, *etc*.), with various combinations of multiple elements and layouts. Complex content and layout, noisy data, font and style variations make automatic document understanding very challenging. For example, to understand text-rich documents such as letters, a system needs to focus almost exclusively on text content, paying attention to a long sequential context, while processing semi-structured documents such as forms requires the system to analyze spatially distributed short words, paying particular attention to the spatial arrangement of the words. Following the success of BERT [3] on NLP tasks, there has been growing interest in developing pretraining methods for document understanding [4, 5, 6]. Pretrained models have achieved state-of-the-art (SoTA) performance across diverse document understanding tasks [7, 8].

Huge training datasets help pretraining models to learn a good representation for downstream tasks. However, we observe three major problems with the current pretraining setup: *(1) documents are composed of semantic regions*. Most of the recent document pretraining works follow BERT and split documents into words. However, unlike the sequence-to-sequence learning in NLP, documents have a hierarchical structure (words form sentences, sentences form a semantic region, and semantic regions form a document). Also, the importance of words and sentences are highly context-dependent, *i.e.*, the same word or sentence may have different importance in a different context. Moreover, current transformer-based document pretraining models suffer from input length constraints. Also, input

35th Conference on Neural Information Processing Systems (NeurIPS 2021).

length becomes a problem for text-rich documents or multi-page documents. *(2) documents are more than words*. The semantic structure of the document is not only determined by the text within it but also the visual features such as table, font size and style, and figure, *etc*. Moreover, the visual appearance of the text within a block are often overlooked. Most of recent BERT-based pre-training works only take the words as input without considering multimodal content and alignment of multimodal information within semantic regions. *(3) documents have spatial layout*. Visual and layout information is critical for document understanding. Recent works encode spatial information via 2D position encoding and model spatial relationships with self-attention, which computes attention weights for long inputs [4, 5]. However, for semi-structured documents, such as forms and receipts, words are more related to their local surroundings. This corresponds strongly with human intuition – when we look at magazines or newspapers, the receptive fields are modulated by our reading order and attention. Based on the above observations, we ask the following question: *Can unified document pretraining benefit all of these different kinds of documents?*

We propose a unified pretraining framework for document understanding, shown in Fig. 1. Our model integrates image information in the pretraining stage by taking advantage of the transformer architecture to learn cross-modal interactions between visual and textual information. To handle textual information, we encode sentences using a hierarchical transformer encoder. The first level of the hierarchical encoder models the formation of the sentences from words. The second level models the formation of the document from sentences. With the help of the hierarchical structure, UDoc learns how words form sentences and how sentences form documents. Meanwhile, it reduces model computation complexity exponentially and increases the number of input words. This also mimics human reading behaviors since the sentence/paragraph is a reasonable unit for people to read and understand—people rarely check the interactions between arbitrary words across different regions in order to understand an article. Convolution has been very successful in the extraction of local features that encode visual and spatial information [9], so we use convolution layers as a more efficient complement to self-attention for addressing local intra-region dependencies in a document image. Meanwhile, self-attention uses all input tokens to generate attention weights for capturing global dependencies. Thus, we combine convolution with self-attention to form a mixed attention mechanism that combines the advantages of the two operations.

We depart from previous vision-language pretraining [10, 11] by extracting both the textual and visual features for each semantic region. We propose a novel gated cross-attentional transformer that enables information exchange between modalities. A visually-rich region (figure, chart, *etc*) may have stronger visual information than textual information. Instead of treating outputs from both modalities identically, we design a gating mechanism that can dynamically control the influence of textual and visual features. This approach enables cross-modal connections and allows for variable highlight the relevant information in visual and textual modality and enables cross-modal connections. During pretraining, the CNN-based visual backbone and multi-layer gated cross-attention encoder are jointly trained in both pretraining and fine-tuning phases.

Our contributions are summarized as follows: (1) We introduce UDoc, a powerful pretraining framework for document understanding. UDoc is capable of learning contextual textual and visual information and cross-modal correlations within a single framework, which leads to better performance. (2) We present Masked Sentence Modeling for language modeling, Visual Contrastive Learning for vision modeling, and Vision-Language Alignment for pretraining. (3) We present extensive experiments and analyses to validate the effectiveness of the proposed UDoc. Extensive experiments and analysis provide useful insights on the effectiveness of the pretraining tasks and show outstanding performance on various downstream tasks.

## 2 Related Work

Self-supervised learning has shown great success in producing generic representations that learn from large-scale unlabeled corpora [3]. Like the development of pretraining in computer vision [12] and NLP [3], there has been a surging interest in self-supervised learning for Vision-Language (VL) tasks [10, 13, 14, 11]. Transformers [3] are the key technology that enables learning contextualized representations from large-scale unlabeled training data. The unique characteristics of document images (spatial layout and multiple elements) distinguish document image pretraining from pretraining works in NLP and VL domains. In the NLP domain, the inputs are pure texts without spatial layouts (bounding boxes). In the VL domain, the inputs are the visual objects and captions. While for

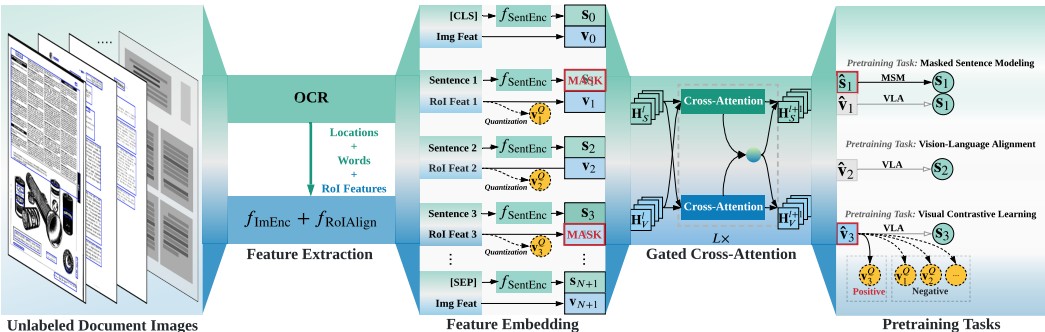

Figure 1: Overview of the proposed approach, UDoc. UDoc first uses a CNN-based visual backbone to learn visual representations. The model then extracts the Region of Interest (RoI) features with OCR bounding boxes and generates a multimodal embedding by combining the textual embedding and position encoding. The transformer-based encoder takes a set of masked multimodal embeddings as input and is pretrained with three pretraining tasks. All the network parameters except those of the textual encoder are jointly trained during both pretraining and fine-tuning phases.

document images, the input elements are spatially distributed, and the visual and textual information co-occur within the semantic regions.

Several recent works have explored pretraining on document images [4, 5, 15]. LayoutLM [4] extends BERT to learn contextualized word representations for document images through multi-task learning. It takes a sequence of OCR words as input during pretraining and incorporates the 2D position embedding as input for each token. However, LayoutLM only considers textual information during pretraining without modeling the alignment between visual and textual information–visual information is only incorporated into the model during the fine-tuning stage. The most recent version, LayoutLMv2 [5], improves on this by incorporating the image encoder into pretraining and jointly training the image encoder along with the BERT model. LayoutLMv2 splits the document image into several parts and concatenates the visual embeddings and text embeddings into a single sequence. Apart from masked language learning (MLM), LayoutLMv2 also considers image-text alignment and image-text matching during pretraining. The most related work to ours is SelfDoc [6], which proposes a multimodal document pretraining framework. It first extracts the document object proposals from pre-trained Faster R-CNN [16] and then applies OCR for each proposal to get the words. It takes the pre-extracted RoI features and sentence embeddings as input, and models the perform learning over the textual and visual information using the cross-modality encoder.

There is a noticeable difference between our proposed method, UDoc, and other concurrent works in document image pretraining. UDoc is a multimodal end-to-end pretraining framework for document images. Unlike the fixed document object detector in [6], the parameters of the image encoder with RoI align, which derive the visual features for semantic regions, are also updated in UDoc. In contrast to [5], our visual features come from the semantic regions instead of splitting the image into fixed regions. Like the object-level semantic elements in natural images, for document images, we represent the typical document layout elements such as paragraph, title, figure, and table as semantic regions. Moreover, to learn the contextualized visual representations, UDoc masks visual information in the latent space and learns contextualized representations by solving a contrastive learning task defined over a quantization of the latent visual embeddings.

## 3 Method

### 3.1 Model Architecture

Fig. 1 illustrates our approach, UDoc, which consists of four components: feature extraction, feature embedding, multi-layer gated cross-attention encoder, and pretraining tasks. Given a document image and the locations of document elements (sentence or RoI), UDoc takes image regions and words that correspond to each document elements as inputs, and extracts their respective embeddings through a visual feature extractor and a sentence encoder. These embeddings are then fed into a transformer-based encoder to learn the cross-modal contextualized embeddings that integrate both visual features and textual features.

In the *feature extraction* step, we first employ an off-the-shelf OCR tool [17] to extract text from a document image $\mathbf{I}$, where the words are grouped into sentences $\mathcal{S} = \{s_1, \ldots, s_N\}$ whose corresponding bounding boxes are $\mathcal{P} = \{p_1, \ldots, p_N\}$. For each sentence bounding box $p_i$, we use a ConvNet-based backbone $f_{\text{ImEnc}}$ and RoI Align [18] $f_{\text{RoIAlign}}$ to extract the pooled RoI features $\boldsymbol{v}_i$. To obtain a *feature embedding*, we extract the sentence embedding $\boldsymbol{s}_i$ for each sentence $s_i$ via a pretrained sentence encoder $f_{\text{SentEnc}}$. Each region's RoI feature $\boldsymbol{v}_i$ is discretized into a finite set of visual representations $\boldsymbol{v}_i^Q \in \mathbf{V}^Q$ via product quantization [19]. The multi-layer *Gated Cross-Attention* encoder takes the position information, masked visual features $\tilde{\mathbf{V}}$ and masked textual features $\tilde{\mathbf{S}}$ as inputs, and then it generates the contextualized multimodal representations ($\mathbf{H}_V^l$ and $\mathbf{H}_S^l$, $l \in [1, L]$) and outputs the predicted features ($\hat{\mathbf{V}}$ and $\hat{\mathbf{S}}$), where $L$ is the number of stacked transformer blocks.

More formally, the pretraining procedure can be decomposed into the following steps:

$$\mathbf{I} \xrightarrow{\text{OCR}} \begin{pmatrix} \mathcal{P} \\ \mathcal{S} \end{pmatrix} \xrightarrow[f_{\text{SentEnc}}]{f_{\text{ImEnc}}+f_{\text{RoIAlign}}} \begin{pmatrix} \mathbf{V}, \mathbf{V}^Q \\ \mathbf{S} \end{pmatrix} \xrightarrow{f_{\text{Mask}}} \begin{pmatrix} \tilde{\mathbf{V}} \\ \tilde{\mathbf{S}} \end{pmatrix} \rightarrow \begin{pmatrix} \mathbf{H}_V^l \\ \mathbf{H}_S^l \end{pmatrix} \rightarrow \begin{pmatrix} \hat{\mathbf{V}} \\ \hat{\mathbf{S}} \end{pmatrix} \rightarrow \mathcal{L}_{\text{Pretraining}} \quad (1)$$

where $f_{\text{Mask}}$ denotes the masking function that randomly masks RoI features and sentence embeddings with the respective probabilities $p_{\text{Mask}}^v$ and $p_{\text{Mask}}^s$. $\mathcal{L}_{\text{Pretraining}}$ is composed of three pretraining tasks: Masked Sentence Modeling (MSM), Visual Contrastive Learning (VCL), and Vision-Language Alignment (VLA). Next, we provide details mentioned in Eq. 1.

**Feature Extraction and Embedding.** Formally, a document image $\mathbf{I} \in \mathbb{R}^{W \times H}$ consists of $N$ regions, where each region's bounding box is characterized by a 6-d vector, as $p_i = \{\frac{x_{\text{LT}}}{W}, \frac{y_{\text{LT}}}{H}, \frac{x_{\text{RB}}}{W}, \frac{y_{\text{RB}}}{H}, \frac{w}{W}, \frac{h}{H}\}$, where $w$ and $h$ are of the width and height the region, $W$ and $H$ are the width and height of $\mathbf{I}$, while $(x_{\text{LT}}, y_{\text{LT}})$ and $(x_{\text{RB}}, y_{\text{RB}})$ denote the coordinates of the top-left and bottom-right corners respectively. The 6-d vector is mapped into a high-dimensional representation via a linear mapping function.

The visual embedding is the sum of the mapped RoI feature and position embedding. Likewise, textual embedding is the sum of sentence embedding and position embedding. We also have different types of segments to distinguish different modalities. The input sequence to the transformer-based encoder starts with a special start element ([CLS] and full visual features), then it is followed by multimodal elements, and it ends with a special ending element ([SEP]+full visual features). For the special elements ([CLS] and [SEP]), the corresponding full visual features are features extracted from the whole input image, by applying $f_{\text{ImEnc}}$ to an RoI covering the whole input image.

**Quantization Module.** Unlike the fixed image encoder in [6], we jointly learn the image encoder in an end-to-end fashion alongside the multimodal model. A visual representation can be learned by predicting the visual features of the masked regions, but it is challenging to predict such features exactly, since they are unconstrained and of continuous representation. To constrain the representation space of the visual features and facilitate the end-to-end learning of image encoder (see Task #2 in Sec. 3.2), we follow [20, 21] and use vector quantization to discretize the visual features $\mathbf{V} = \{\boldsymbol{v}_1, \ldots, \boldsymbol{v}_N\}$ into a finite set of representations $\mathbf{V}^Q = \{\boldsymbol{v}_1^Q, \ldots, \boldsymbol{v}_N^Q\}$. Specifically, we define latent embedding spaces $\boldsymbol{e} \in \mathbb{R}^{C \times E}$, where $C$ is the number of codebooks, and $E$ is the number of entries for each codebook. For each $\boldsymbol{v}_i$, we first map it to logits $\boldsymbol{v}_i^\ell \in \mathbb{R}^{C \times E}$, and calculate the probability for the $j$-th codebook entry in $i$-th group as $p_{c,e} = \exp((v_{c,e}^\ell + g_e)/\tau) / \sum_{k=1}^E \exp((v_{c,k}^\ell + g_k)/\tau)$, where $\tau$ is a non-negative temperature, $g_{1:E}$ are i.i.d samples drawn from Gumbel(0,1) distribution. During the forward pass, we choose one entry vector from each codebook by $\tilde{\boldsymbol{e}}_i \sim \text{argmax}_e p_{c,e}$ and generate the quantized representation $\boldsymbol{v}_i^Q$ by a concatenation of $\{\tilde{\boldsymbol{e}}_1, \ldots, \tilde{\boldsymbol{e}}_G\}$ which is then followed by a linear transformation. During the backward pass, the gradients are computed through a Gumbel-Softmax estimator [22].

**Gated Cross-Attention.** To model the interactions among multimodal inputs, we introduce a multimodal transformer with gated cross-attention to model the cross-modality relationships. Let $\mathbf{H}_m^{l+1}$ be output features at the $l$-th layer for one modality $m$, and let $n$ be another modality ($m, n \in \{V, S\}$). We obtain the features at $(l+1)$-th layer as:

$$\mathbf{H}_m^{l+1} = f_{\text{LN}}\Big( f_{\text{LN}}\big(\mathbf{H}_m^l + f_{\text{Cross-Att}}^l(\mathbf{H}_m^l | \mathbf{H}_n^l)\big) + f_{\text{FF}}^l\big(f_{\text{LN}}(\mathbf{H}_m^l + f_{\text{Cross-Att}}^l(\mathbf{H}_m^l | \mathbf{H}_n^l))\big)\Big) \quad (2)$$

where $f_{\text{LN}}$ denotes layer normalization [23]. The feed-forward sub-layer $f_{\text{FF}}$ in Eq. 2 is further composed of two fully-connected sub-layers, both wrapped in residual adds and $f_{\text{LN}}$.

The core part of Eq. 2 is the cross-attention $f_{\text{Cross-Att}}(\cdot)$. Given the intermediate representations $\mathbf{H}_m^l$ and $\mathbf{H}_n^l$, the cross-attention output for modality $m$ is computed as:

$$f_{\text{Cross-Att}}(\mathbf{H}_m^l|\mathbf{H}_n^l) = [\text{Cross-Att}^1(\mathbf{H}_m^l|\mathbf{H}_n^l); \ldots; \text{Cross-Att}^h(\mathbf{H}_m^l|\mathbf{H}_n^l)]\boldsymbol{U} \tag{3}$$

$$\text{Cross-Att}^i(\mathbf{H}_m^l|\mathbf{H}_n^l) = \text{softmax}\left(f_q^i(\mathbf{H}_m^l)f_k^i(\mathbf{H}_n^l)^T/\sqrt{d}\right)f_v^i(\mathbf{H}_n^l) \tag{4}$$

where $f_q^i(\mathbf{H}_m^l)$, $f_k^i(\mathbf{H}_n^l)$, and $f_v^i(\mathbf{H}_n^l)$ are the *query*, *key*, and *value* calculated by linear mapping layers for the $i$-th head. $d$ is the model dimension, $h$ is the number of heads, and $\boldsymbol{U}$ is the weight matrix that combines the outputs of the heads.

Considering the substantial diversity of document images and the different information needs of differing document types, we use a gating mechanism [24] to dynamically weight the outputs of the visual and textual branches. Specifically, we feed the concatenated the visual and textual features to a non-linear network $f_{\text{Gate}}([\mathbf{H}_m^{l+1}; \mathbf{H}_n^{l+1}])$, which generates the modality-specific attention weights $\alpha_m^l$ and $\alpha_n^l$, and returns the weights separately to their respective modality-specific branches to perform element-wise products. We multiply the features for modality $m$ with its modality-specific attention weight, and compute the updated feature as: $\mathbf{H}_m^{l+1} = \mathbf{H}_m^{l+1}(1 + \alpha_m^l)$, same that for modality $n$.

## 3.2 Training Tasks and Objectives

The full pretraining objective of UDoc (right block in Fig. 1) is defined as: $\mathcal{L}_{\text{Pretraining}} = \mathcal{L}_{\text{MSM}} + \mathcal{L}_{\text{VCL}} + \mathcal{L}_{\text{VLA}}$. In the rest of this section, we describe each task in detail.

**Task #1**: **Masked Sentence Modeling.** This task is similar to the MLM task utilized in BERT. The key difference is that we mask sentences instead of tokens. During pretraining, each sentence and RoI of the input document is randomly and independently masked. For the masked sentence, its token is replaced with a special sentence of [MASK]. The model is trained to predict the masked sentence feature, based on the unmasked words and the visual features. The goal is to predict the masked sentence embeddings based on the contextual information from the surrounding sentences and image regions, by minimizing the smooth L1 loss [16]:

$$\mathcal{L}_{\text{MSM}}(\Theta) = \sum_i \text{smooth}_{L_1}(\boldsymbol{s}_i - f_{\text{UDoc}}(\boldsymbol{s}_i|\boldsymbol{s}_{\backslash i}, \tilde{\mathbf{V}})) \tag{5}$$

where $\Theta$ is the trainable parameters and $f_{\text{UDoc}}(.)$ outputs the unmasked textual feature, $\boldsymbol{s}_{\backslash i}$ is the surrounding features for the $i$-th input, $\tilde{\mathbf{V}}$ are the image features with random masking.

**Task #2**: **Visual Contrastive Learning.** We learn visual feature representations by solving a visual contrastive learning task which requires estimating the true quantized latent RoI representation. Given a prediction $\hat{\boldsymbol{v}}_i \in \hat{\boldsymbol{V}}$ for the masked RoI $\tilde{\boldsymbol{v}}_i \in \tilde{\mathbf{V}}$, the model needs to estimate the positive quantized representation $\boldsymbol{v}_i^Q$ in a set of quantized candidate representations $\mathbf{V}^Q$. Good representations are learned by maximizing the agreement between output representation and quantized representation of the same RoIs as follows:

$$\mathcal{L}_{\text{VCL}}(\Theta) = -\sum_{\tilde{\boldsymbol{v}}_i \in \tilde{\mathbf{V}}} \left( \log \frac{\exp(\text{sim}(\hat{\boldsymbol{v}}_i, \boldsymbol{v}_i^Q)/\kappa)}{\sum_{\boldsymbol{v}_j^Q} \exp(\text{sim}(\hat{\boldsymbol{v}}_i, \boldsymbol{v}_j^Q)/\kappa)} \right) + \lambda \frac{1}{CE} \sum_{c=1}^{C} \sum_{e=1}^{E} p_{c,e} \log p_{c,e} \tag{6}$$

where $\text{sim}(\cdot, \cdot)$ computes the cosine similarity between two vectors, $\lambda$ is a hyperparameter, and $\kappa$ is a temperature scalar. The second term encourages the model to use the codebook entries more equally.

**Task #3**: **Vision-Language Alignment.** To enforce the alignment among different modalities, we explicitly encourage alignment between words and image regions via similarity-preserving knowledge distillation [25]. Note that, unlike the text-image alignment in LayoutLMv2 [5] which splits the image into four regions and predicts whether the given word is covered or not on the image side, we align the image and text belonging to the same region. The goal is to minimize the differences

between the pairwise similarities of sentence embeddings and the pairwise similarities of image region features:

$$\mathcal{L}_{\text{VLA}}(\Theta) = \frac{1}{N \times N}||f_{\text{Norm}}(\mathbf{S} \cdot \mathbf{S}^\top) - f_{\text{Norm}}(\mathbf{H}_V^L \cdot \mathbf{H}_V^{L\top})||_F^2 \tag{7}$$

where $\mathbf{S}$ is the unmasked input sentence embeddings, $\mathbf{H}_V^L$ is the mapped visual representations of the final layer, $|| \cdot ||_F$ is the Frobenius norm, and $f_{\text{Norm}}$ performs L2 normalization.

## 4 Experiment

### 4.1 Pretraining UDoc

**Pretraining corpus.** We build our pretraining corpus based on IIT-CDIP Test Collection 1.0 [26], which contains more than 11M scanned document images. To differentiate pretraining from finetuning, we filter out the document images of RVL-CDIP [8] from IIT-CDIP since it is a subset of IIT-CDIP, and sample 1M document images as our pretraining corpus.

Table 1 shows the dataset statistics. IIT-CDIP only provides the OCR texts in XML format. We extract words and their locations by applying EasyOCR [17] on document images. As shown in Fig. 3 (a), EasyOCR provides two kinds of output modes: non-paragraph and paragraph. The paragraph mode groups the non-paragraph results into text regions. We think document image pretraining should be treated differently than sequence-based pretraining in NLP, since the words in the document (2D) are arranged according to spatial layouts, while the words in

Table 1: Comparison of the datasets used for pretraining and finetuning process. 'Box', 'Label', and 'Text' indicate the availability of location, label and text annotations for document entities. 'Tag' denotes the document class label availability.

| Dataset | Type | Size | Box | Label | Text | Tag |
|---|---|---|---|---|---|---|
| IIT-CDIP [26] | Misc | 11M | ✗ | ✗ | ✓ | ✗ |
| RVL-CDIP [8] | Misc | 400K | ✗ | ✗ | ✗ | ✓ |
| CORD [7] | Receipt | 1K | ✓ | ✓ | ✓ | ✗ |
| FUNSD [27] | Form | 0.2K | ✓ | ✓ | ✓ | ✗ |
| PubLayNet [28] | Article | 347K | ✓ | ✓ | ✗ | ✗ |

NLP corpora are sequential (1D). Considering the special characteristics of documents (complex layout, multi-pages) and the limited input length of BERT models, it is not intuitive to formulate the input at the word level. Hence, we adopt the paragraph-level outputs as the basic input elements since textual regions provide semantically more meaningful information than independent words.

There are some advantages to our design: (1) the region-level design hierarchically encodes document elements and this facilitates the modeling of latent relationships at the region level which has higher-level semantics than the word level. (2) the hierarchical encoding also overcomes the input size limitation of word-level BERT-based models [4, 5]. Fig. 2 shows the distribution of words per region on RVL-CDIP. It can be seen that even though we consider region-level input, for some semi-structured documents, single-words dominate the inputs; this somehow forces UDoc to pay attention to word-level inputs. Unlike MLM that predicts the masked word, UDoc predicts the textual embedding of the masked input with MSM.

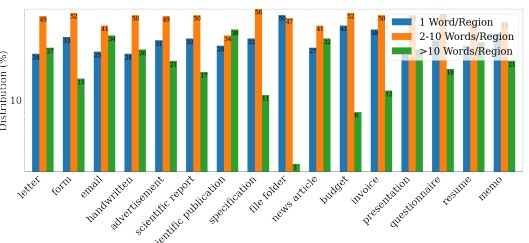

Figure 2: Distribution of words per region on RVL-CDIP according to the categories.

**Pretraining setting.** We initialize the sentence encoder $f_{\text{SentEnc}}$ with BERT-NLI-STSb-base [29] pretrained for NLI [30] and STS-B [31]. The ResNet-50 backbone in the image encoder is pretrained on the PubLayNet training set [28]. All the parameters (except $f_{\text{SentEnc}}$ and $f_{\text{ImEnc}}$) are randomly initialized. During pretraining, we freeze the parameters of $f_{\text{SentEnc}}$ and jointly train the visual encoder and multi-modal UDoc model in an end-to-end fashion. Such an end-to-end training allows the ConvNet and Transformer to realize their full potentials in spatial and sequence modeling for pretraining. UDoc contains 12 layers of gated cross-attention transformer blocks. We set the hidden size to 768 and the number of heads to 12, the maximum number of regions $N$ to 64, and the maximum input sequence length for $f_{\text{SentEnc}}$ to 512. The pretraining is conducted on 8 NVIDIA Tesla

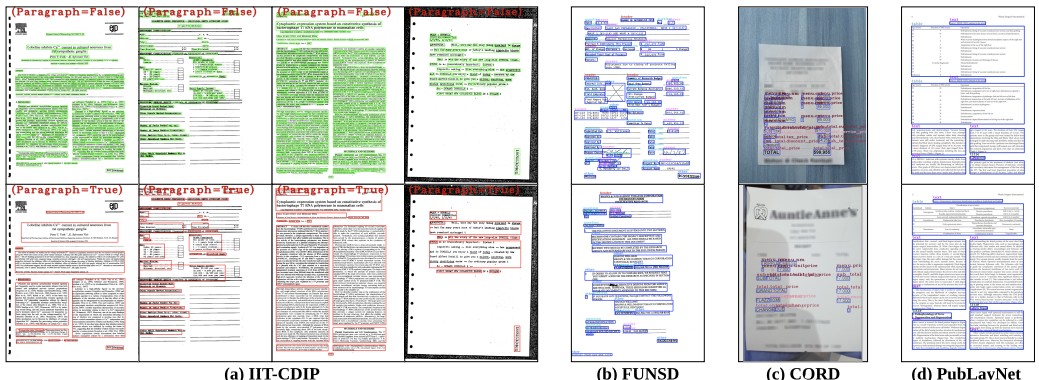

**Figure 3:** Document image samples. The boxes in red/green are OCR bounding boxes obtained with/without paragraph mode, while the boxes in blue are officially-provided bounding boxes.

V100 32GB GPUs with a batch size of 64. It is trained with Adam optimizer [32], with an initial learning rate of $10^{-5}$, weight decay of $10^{-4}$, and learning rate warmup in the first 20% iterations.

To learn a useful multimodal representation, random masking is applied to both textual and visual inputs. For MSM, we set the mask probability $p_{\text{Mask}}^s$ for input sentences to 15%. 80% among the masked sentences are replaced by special sentence [CLS, MASK, SEP], while 10% sentences are replaced by random sentences sampled from other documents, and 10% remains unchanged. For VCL, the $\lambda$ is set to 0.1, $\kappa$ is set to 0.1, the mask probability $p_{\text{Mask}}^v$ is set to 7.5% and the masked RoI features are filled with zeros. The temperature $\tau$ is annealed from 2.0 to 0.5 by a factor of of 0.999995 at every iteration. We select the pretraining checkpoint with the lowest $\mathcal{L}_{\text{Pretraining}}$ for finetuning stage.

## 4.2 Finetuning Tasks

**Form Understanding.** Form understanding requires the model to predict the label for each semantic entity. We use FUNSD [27] as the evaluation dataset. It contains 149/50 training/testing images. Fig. 3 (b) shows a sample from FUNSD. Each semantic entity comprises a list of words, a label, and a bounding box. The officially-provided OCR texts and bounding boxes are used during training and testing. We take the semantic entities as input and feed the concatenated visual and textual output representations to a classifier. We apply cross-entropy loss for finetuning. The model is finetuned for 100 epochs with a learning rate of $10^{-5}$ and batch size of 16. All the parameters except $f_{\text{SentEnc}}$ are trained. One of *question*, *answer*, *header* or *other* is predicted for each semantic entity. We use entity-level F1 score as the evaluation metric.

**Receipt Understanding.** Receipt understanding requires the model to recognize a list of text lines with bounding boxes. The performance on this task is evaluated on CORD [7] dataset. It contains 626/247 receipts for training/testing. The receipts are labeled with 30 types of entities under 4 categories: *company*, *date*, *address*, and *total*. Like FUNSD, we feed the concatenated visual and textual output representations to the classifier. The model is finetuned for 200 epochs with a batch size of 16 and a learning rate of $10^{-5}$. The evaluation metric is entity-level F1 score.

**Document Classification.** Document classification involves predicting the category for each document image. We use RVL-CDIP [8] as the target dataset. It consists of 320K/40K/40K training/validation/testing images under 16 categories. The OCR words and bounding boxes are extracted by EacyOCR. To fine-tune UDoc on RVL-CDIP, we compute the overall representation as an element-wise product between the visual and textual representations averaged from all sentences/regions, and learn a classifier on top of the overall representation with cross-entropy loss. We fine-tune the model for 30 epochs with a batch size of 64 and a learning rate of $10^{-5}$. Classification accuracy over 16 categories is used to measure model performance.

**Document Object Detection.** Document object detection involves decomposing a document image into semantic units. We evaluate the effectiveness of our pretrained visual backbone on PubLayNet [28]. As shown Fig. 3 (d), the documents in PubLayNet are scientific articles. PubLayNet consists of 336K/11K training/validation images with six category labels (*text*, *title*, *list*, *figure*, and

*table*). We train Faster-RCNN (F-RCNN) using Detectron2 [33] and initialize the visual backbone with the pretrained ResNet-50 from UDoc. The model is trained for 180k iterations with a base learning rate of 0.01 and a batch size of 8. Mean average precision (MAP) @ intersection over union (IOU) [0.50:0.95] of bounding boxes is used to measure the performance.

### 4.3 Results and Discussion

**The importance of multimodal learning.** To study the effect of multimodal learning, we experiment in three different settings (1) Vision only (V): this setting omits the textual components of UDoc and adopts multilayer self-attention transformer to learn the visual representation. (2) Language only (L): this setting omits the visual encoder and keeps only the textual components. (3) Vision-Language (V+L): this setting considers both vision and language information. We first train three settings without pretraining. Table 2 shows consistent improvement across tasks for V+L over the single-stream baselines (V or L). This demonstrates that our UDoc model is able to learn important visual-linguistic relationships that benefit downstream tasks even without pretraining.

In Table 2, we find that visual information dominates the performance of document classification, while language information contributes a lot to form understanding and receipt understanding. The results also indicate that different document tasks rely on different information. For document entity recognition tasks, language information is more important than visual features. As can be seen in Fig. 3 (b) and (c), entity recognition is more word-oriented. On the other hand, document classification is more focused on global-level understanding. As a result, visual and layout information contribute a lot to the final prediction of the document classification model. This matches well with the innate abilities of humans to distinguish between document types without fully understanding the words. We also observe that gated cross-attention (V+L) achieves a better performance than the non-gated version (V+L$^\sharp$), as its gating mechanism can learn to adaptively determine how much each modality contributes to the output features.

**Effect of pretraining tasks.** We analyze the effectiveness of different pretraining settings through ablation studies over FUNSD, CORD, and RVL-CDIP, which are representative document benchmarks. Table 2 ablates the key design choices in pretraining UDoc. For experimental efficiency, UDoc models evaluated here are trained with 5 epochs on 300k training corpus. Overall, the pretraining of UDoc consistently improves the performance over all three downstream tasks. The improvement gains vary among different tasks.

Table 2: Experimental results and comparison on FUNSD, CORD, and RVL-CDIP test sets.

| | | | Pretraining | | | | FUNSD | CORD | RVL-CDIP |
|---|---|---|---|---|---|---|---|---|---|
| Enable | #Data | Modality | Max #Words | #Param. | Tasks | Epoch | F1 | F1 | Accuracy |
| ✗ | – | V | – | 85M | – | – | 77.49 | 57.08 | 91.35 |
| | – | L | – | 153M | – | – | 78.46 | 71.52 | 86.82 |
| | – | V+L$^\sharp$ | – | 255M | – | – | 80.60 | 95.98 | 92.76 |
| | – | V+L | – | 267M | – | – | 83.34 | 96.59 | 92.93 |
| ✓ | 300K | V+L | $64 \times 512$ | 270M | MSM + MVM | 5 | 84.37 | 97.44 | 93.10 |
| | 300K | V+L | $64 \times 512$ | 272M | MSM + VCL | 5 | 86.87 | 98.70 | 93.59 |
| | 300K | V+L | $64 \times 512$ | 272M | MSM + VCL + VLA | 5 | 87.38 | 98.75 | 93.92 |
| | 300K | V+L | $64 \times 512$ | 274M | MSM + VCL + VLA + REL | 5 | 87.20 | 98.13 | 93.64 |

We first establish two baselines: MSM+MVM in Table 2 indicates the combination of masked sentence learning and masked visual feature prediction. Similar to MSM, for MVM, we freeze the visual backbone and perform masked visual feature prediction via RoI-feature regression. MSM+VCL jointly trains the visual backbone end-to-end with contrastive learning. As shown in Table 2, MSM+MVM achieves better results than the model without pretraining. Furthermore, when combining VCL together with MSM, consistent performance gains are observed across all the benchmarks. Among the three finetuning tasks, the improvements on FUNSD and CORD are bigger than on RVL-CDIP. We think the local context modeling capability of the ConvNet-based image encoder brings more benefits to entity recognition, since entities are heavily linked and correlated to their local surroundings. When MSM, VCL, and VLA are jointly trained, we observe further performance gains across all the benchmarks. For VCL, instead of sampling the negatives from the same input document, we also try including the negative samples from other document images of the same batch. However, we find that sampling negatives from the entire batch of document images hurt the performance. This is likely because the negatives from other document images are easy to distinguish from each other. We

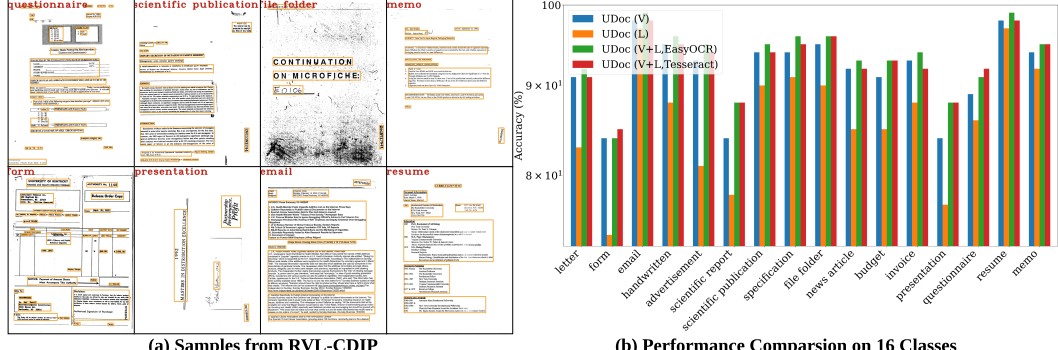

| (a) Samples from RVL-CDIP | (b) Performance Comparsion on 16 Classes |

Figure 4: For (a) we show the samples from RVL-CDIP. The boxes in orange color are grouped OCR bounding boxes. For (b) we plot the accuracies on 16 classes achieved by different models that are represented by different colors in the bar chart.

also consider the image-text matching task (Rel) [5], and combine Rel with MSM+VCL+VLA. It hurts the performance of all three downstream tasks. We conjecture that the image-text matching task introduces mismatched pairs of image and OCR texts as negative examples that potentially hamper the training of other tasks.

**What if Masked Language Modeling is included?**    To study the feasibility of that, we consider MLM during pretraining. Since the number of words may be very large, we select the tokens by randomly applying a sliding window (window size 128) across all sequenced OCR words. Each word is formulated as a single-word sentence ([CLS] [Token] [SEP]). We randomly mask 15% of those sampled words ([CLS] [MASK] [SEP]) and concatenate them along with the region-based inputs. During pretraining, we add the word prediction head on top of UDoc and predict the masked words. We conduct finetuning experiments on entity recognition tasks (FUNSD and CORD), and find that such a direct combination hurts the performance: FUNSD: 87.38 (UDoc) vs. 83.76↓ (UDoc+MLM), CORD: 98.75 (UDoc) vs. 98.63↓ (UDoc+MLM). There are consistent performance drops from adding MLM. One possible reason is that the RoI features extracted by token bounding boxes might not be discriminative enough due to the tiny word-level bounding boxes.

Table 3: Comparison with state-of-the-art methods. The symbol ‡ implies using Google OCR engine.

| Method | Pretraining | | | | | | FUNSD | CORD | RVL-CDIP |
| | Source | #Data | Scale | Max #Words | Modality | #Param. | F1 | F1 | Accuracy |
| --- | --- | --- | --- | --- | --- | --- | --- | --- | --- |
| BERT$_{BASE}$ [5] | – | – | Word | 512 | L | 110M | 60.26 | 89.68 | 89.81 |
| BERT$_{LARGE}$ [5] | – | – | Word | 512 | L | 340M | 65.63 | 90.25 | 89.92 |
| LayoutLM$_{BASE}$ [5] | IIT-CDIP | 11M | Word | 512 | L | 113M | 78.66 | 94.72 | 94.42 |
| LayoutLM$_{LARGE}$ [5] | IIT-CDIP | 11M | Word | 512 | L | 343M | 78.95 | 94.93 | 94.43 |
| LayoutLMv2$_{BASE}$ [5] | IIT-CDIP | 11M | Word | 512 | V+L | 200M | 82.76 | 94.95 | 95.25 |
| LayoutLMv2$_{LARGE}$ [5] | IIT-CDIP | 11M | Word | 512 | V+L | 426M | 84.20 | 96.01 | 95.64 |
| SelfDoc [6] | RVL-CDIP | 320K | Region | 50×512 | V+L | – | 83.36 | – | 92.81 |
| SelfDoc+VGG-16 [6] | RVL-CDIP | 320K | Region | 50×512 | V+L | – | – | – | 93.81 |
| TILT-Base [34] | RVL-CDIP+ | 1.1M | Word | 512 | V+L | 230M | – | 95.11 | 95.25 |
| TILT-Large [34] | RVL-CDIP+ | 1.1M | Word | 512 | V+L | 780M | – | 96.33 | 95.52 |
| UDoc | IIT-CDIP | 1M | Region | 64×512 | V+L | 272M | 87.96 | 98.85 | 93.96 |
| UDoc* | IIT-CDIP | 1M | Region | 64×512 | V+L | 272M | 87.93 | 98.94 | 95.05‡ |

**Performance Comparison with SoTA.**    We further pretrain UDoc on 1M document images with 5 epochs and report the finetuning results in Table 3. UDoc outperforms previous models on FUNSD and CORD, by a significantly large margin, demonstrating that our proposed approach is highly effective, partially due to the end-to-end training of the image encoder that improves the semantic alignments between images and texts. Note that UDoc is pretrained on a subset of IIT-CDIP (1M document images), which is considerably less than the 11M document images used in LayoutLM [4] and LayoutLMv2 [5]. TILT [34] builds a 1.1M pretraining corpus by combining RVL-CDIP, UCSF Industry Documents Library, and Common Crawl. UDoc also achieves promising results on document classification. Note that both LayoutLM v2 and TILT use Microsoft OCR, which is a commercial service with a stronger OCR performance than EasyOCR, which is used in our experiments. We find that OCR plays a key role in document classification performance. As shown in Fig. 4, UDoc

performs the best on the '*email*' category but worst on the '*form*' category. We also report the results with different OCR engines: 93.42 (Tesseract [35]) vs. 93.96 (EasyOCR [17]) vs. 94.10 (Google OCR [36]). UDoc with EasyOCR achieves a better performance than with Tesseract since EasyOCR is powered by an advanced neural network, while Tesseract is based on less sophisticated techniques. Since different tasks require task-specific input embeddings to perform well, instead of finetuning the sentence encoder during pretraining, we explore unfreezing the sentence encoder during the finetuning stage (named as UDoc$^*$) and report the results in Table 3. Unsurprisingly, we see performance improvements on several downstream applications. *E.g.*, RVL-CDIP: 93.96 (UDoc) vs. 95.05↑ (UDoc$^*$). However, this also makes the training more challenging in terms of computational resources and training time.

**Effect of visual backbone.** Additionally, we apply the trained visual backbone to document object detection on PubLayNet. The performance of the F-RCNN on the validation set is depicted in Table 4. To better compare, we establish two F-RCNN models with: (1) backbone initialized with ResNet-50 pretrained on ImageNet; (2) backbone initialized from UDoc's pretrained visual backbone.

Table 4: MAP @ IOU [0.50:0.95] of the document detection models on PubLayNet dev set.

| Method | Text | Title | List | Table | Figure | mAP |
|---|---|---|---|---|---|---|
| F-RCNN (ResNet-101) [28] | 91.0 | 82.6 | 88.3 | 95.4 | 93.7 | 90.0 |
| M-RCNN (ResNet-101) [28] | 91.6 | 84.0 | 88.6 | 96.0 | 94.9 | 90.7 |
| F-RCNN (ResNet-50) | 92.2 | 84.4 | 89.5 | 96.5 | 94.5 | 91.4 |
| F-RCNN (UDoc, ResNet-50) | 93.9 | 88.5 | 93.7 | 97.3 | 96.4 | 93.9 |

It can be seen that our pretrained backbone outperforms ImageNet-pretrained backbones. By leveraging UDoc, we can train different variants of the visual backbone and apply them to document-specific downstream applications, without relying on incompatible pretrained backbones from other domains (*e.g.*, natural image). Moreover, the visual backbone of UDoc does not require any custom layers, and thus any ConvNet architecture can be used in place of ResNet.

## 5   Conclusion, Limitations, and Future Works

We develop UDoc, a unified pretraining framework for document understanding. Our model introduces a novel joint training framework that effectively exploits the visual and textual information during pretraining and finetuning. We evaluate the UDoc comprehensively on three downstream tasks: form understanding, receipt understanding, and document image classification. Extensive empirical analysis demonstrates that the pretraining procedure can take advantage of multimodal inputs. Also, it can effectively aggregate and align visual and textual information of document images with the proxy tasks. This work has a broader impact on document applications. By finetuning the pretrained UDoc on task-specific data, document processing systems can provide better results and reduce the expensive data annotations costs. In terms of negative social impact, the document images used for pretraining may contain sensitive information and therefore the models trained on such data may inappropriately leak some private information. To address the privacy leakage, it is worthwhile to explore the combination of privacy-preserving learning and self-supervised learning.

There are interesting short- and long-term research directions for UDoc: (1) we freeze the sentence encoder during pretraining and fine-tuning phases due to computational constraints. A better document representation can be learned by jointly training the sentence encoder, visual backbone and cross-attention encoder in a completely end-to-end fashion. (2) Although impressive performance has been achieved in document entity recognition tasks such as form and receipt understanding, the classification accuracy on semi-structured documents such as forms is still inferior to that of rich-text documents. It is possible to devise a better method to model the spatial relationship among words. (3) An interesting direction is to extend UDoc to multipage/multilingual document pretraining. Additionally, there exist many text-based labeled document datasets in the NLP domain, such as document summarization. Can we transfer the knowledge learned from the text-based document domain to the image-based document domain? How to unify the pretraining of the pure-text document (1D) and image-based document (2D) in a single framework is also worth to try. Lastly, the use of different OCR tools is one of the major sources of inconsistency among the existing document pretraining works. It is worthwhile and essential to build standardized pretraining document image datasets with preprovided OCR results. In addition to scanned documents, using digital PDF as part of the pretraining data is a direction worth exploring since it provides rich metadata which could be beneficial for multimodal learning.

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
