# OpenReview forum: "UniDoc: Unified Pretraining Framework for Document Understanding"
_NeurIPS.cc/2021/Conference — NeurIPS 2021 Poster_

### Official Review · Reviewer_9qdW · 2021-07-07

**Rating:** 6
**Confidence:** 4

**Summary:**

This paper presents UniDoc as a multimodal pre-trained model for document understanding. UniDoc consists of 3 layers: a feature extraction layer, which extracts sentence embeddings and region embeddings from the original document; a fusion layer, which fuses textual and visual information based on a gated cross-attention mechanism; a task layer, which uses 3 tasks as pre-training objectives to optimize model parameters. Evaluations are conducted on 4 downstream tasks, including Form Understanding, Receipt Understanding, Document Classification and Document Object Detection. Comparing to several SOTA baselines, UniDoc performs better on Form Understanding and Receipt Understanding, but worse on Document Classification. The paper also includes some ablation studies to investigate the effects of V/L/V+L, pre-training tasks and visual backbones, and some interesting future directions.

**Limitations And Societal Impact:**

As the paper said: "...the document images used for pretraining may contain sensitive information and therefore the models trained on such data may inappropriately leak some private information...".

**Main Review:**

The strengths of the papers: (1) A multimodal pre-trained model for document understanding, where both visual regions and texts extracted from visual regions are used as inputs; (2) 3 specific pre-training tasks designed document understanding are proposed, which show strong gains on some of the downstream tasks.

The weaknesses of the paper: (1) The effects of some factors are not fully discussed (listed in the question part); (2) The result on RVL-CDIP is not SOTA. The paper said that the potential reason could be using a weaker OCR, but not prove it directly.

In general, I think this is a good paper for document understanding. I give 6 this time but would like to change the score if the questions are addressed in the rebuttal period.

Questions: (1) What is the effect of the gated mechanism used in cross-attention? What the results look like if the gate is removed? (2) Can we replace the gated cross attention layer with a normal Transformer (a single-stream model)? (3) Is it possible to include Masked Language Models as well? If so, will the results become better with this typical pre-training task? (4) Are discrete representations indispensable? For example in Figure 1, can we use the RoI features from other documents in the same batch as negative examples for contrastive learning?

**Time Spent Reviewing:**

2.5 hours

---

> ### Author Response · Authors · 2021-08-10
> **Response to Reviewer 9qdW**
>
>
> We thank the reviewer for the encouraging words and helpful comments.
>
> ### **Q: Effect of the gated mechanism used in cross-attention.**
> Thanks for pointing this out.
>
> During our initial exploration of the cross-attention component, we did consider the two possible designs: gated vs. not-gated. We select the gating mechanism based on the following motivations and findings:
>
> **[1]** Since we formulate the input to UniDoc as a set of regions (visual and textual features), we design our proposed gated cross-attention based on the idea of gating mechanism for adaptively selecting the parts of the input that are more likely to contribute to the downstream document applications.
>
> **[2]** During our initial framework exploration, we train the different settings from scratch on downstream tasks (as shown in Table 2, without pretraining), and found that gated cross-attention achieved better performance than no-gated version, since it can learn a gate-activation pattern that adaptively determines how much each modality contributes to the output features. Specifically, the performance comparison between gated vs. not-gated is as follows: FUNSD (83.34 vs. 80.60$\downarrow$), CORD (96.59 vs. 95.98$\downarrow$), and RVL-CDIP (92.93 vs. 92.76$\downarrow$).
>
> ### **Q: Replace gated cross-attention with single-stream model.**
>
> Great comment. As suggested, we compare UniDoc (gated cross-attention) with UniDoc (single self-attention). Specifically, we concatenate the visual and textual features and feed them to a 12-layer self-attention (12 heads) transformers. We train the model from scratch (without pretraining) and report the results on RVL-CDIP: 92.93 (Gated Cross-Attention) vs. 92.11$\downarrow$ (Single Self-Attention). This shows that gated cross-attention can learn better representation for each modality by conditioning on one another.
>
> ### **Q: Result on RVL-CDIP and Effectiveness of better OCR.**
>
> As suggested, to illustrate the effectiveness of different OCR tools, we conduct additional ablation experiments: Tesseract OCR vs. Easy OCR vs. Google OCR, and the results: Tesseract OCR (93.42$\downarrow$) vs. Easy OCR (93.96) vs. Google OCR. (94.02$\uparrow$). Note that, due to time and resource limitations, we only extract the RVL-CIDP with Google OCR, without extracting full IIT-CDIP or pretraining UniDoc with new Google OCR results.
>
> We further explore the finetuning of the entire framework (unfreeze sentence encoder) on RVL-CDIP (with Google OCR); until now, the performance on RVL-CDIP improves (93.96 $\rightarrow$ 95.05$\uparrow$). We will extract the OCR results via Google OCR on the full IIT-CDIP document images and include the relevant experiments (pretraining with better OCR) in our revised paper.
>
> Lastly, we have also witnessed that the use of different preprocessing (OCR) tools is one of the major sources of inconsistency among the existing document pretraining works. The major reason that different OCR tools are used by existing works is that the pretraining corpus, such as IIT-CDIP, only provides the document images (without bounding boxes). We think it is worthwhile and essential to build a better pretraining document image dataset and release the preprocessed OCR results that are beneficial to the community.
>
> ### **Q: Include masked language modeling.**
>
> Nice question. It is true that masked language modeling has been successfully applied to previous vision/vision-language/document pretraining works, such as LayoutLM and TILT. We answer this question from two aspects:
>
> **[1]** Firstly, we think, unlike the document (1D) in the NLP domain, the document images in our setting are more than just words. Although most of the existing works treat words as the input element for this task, we think token-level input may not be suitable for documents since documents can be multimodal, multipage, and multilingual. For instance, a NeurIPS paper has multiple pages (>8) and thousands of words. Modeling such a long and rich-text document at word-level is not feasible with a BERT model due to the length limitation (e.g., 512) of BERT. That is the main reason we did not consider token-level during pretraining. Although we do not incorporate masked token learning into UniDoc, we found that a large portion of input sentences are single-word (please refer to the Table shown in the response to Reviewer xpLb). Thus, word-level modeling has already somehow been considered by the token-level modeling of our model during pretraining (instead of predicting the masked words, UniDoc learns to predict the masked embeddings.)
>
> **[2]** Nevertheless, we agree that it is worth trying masked language learning in the pretraining. To combine region-level and token-level into a single pretraining framework, we come up with the following design: a) sample 128 tokens according to a random sliding window and treat each token as a sentence [CLS] [Token] [SEP], and concatenate those masked (masking probability is 15\%) token-level inputs with region-level inputs, and feed them into UniDoc. b) add an additional word prediction head on top of those token-level outputs; (c) train such multi-granularity (regions+tokens) UniDoc on 300k pretraining corpus (same as Table 2), and finetune the model for the downstream applications. The results are as follows: FUNSD: 87.38 (UniDoc) vs. 83.76$\downarrow$ (UniDoc+MLM), CORD: 98.75 (UniDoc) vs. 98.63$\downarrow$ (UniDoc+MLM). There are consistent performance drops from adding masked token learning. One possible reason is that the bounding boxes for the tokens are so small to extract sufficiently discriminative and useful RoI features, therefore introducing some undesirable noise to the model.
>
> ### **Q: Negative examples from other documents in the same batch for contrastive learning.**
>
> We did consider the negative samples from other document images of the same batch. However, we found that sampling negatives from the entire batch of document images hurt the performance. We conduct the pretraining (sampling negatives from the same batch) on 300k corpus, and report the finetuning results (using negatives from same document vs. negatives from same batch) as follows: FUNSD (87.38 vs. 84.81$\downarrow$); CORD (98.75 vs. 96.95$\downarrow$). This is likely because the negatives from other document images are easy to distinguish from each other. In the revised version, we will include the experimental results for the setting with negative sampling from the same batch.

---

> > ### Comment · Reviewer_9qdW · 2021-08-23
> > **thank you for the rebuttal**
> >
> > Thank you for your detailed answers to my questions, which addressed most of them! I like these newly added experiments and findings. Please keep them in the revised version of the paper.

---

> > > ### Author Response · Authors · 2021-08-27
> > > **Thank you!**
> > >
> > > Thank you for your helpful comments and support.

---

### Official Review · Reviewer_Zk5e · 2021-07-11

**Rating:** 6
**Confidence:** 3

**Summary:**

This paper proposes a multi-modal pretraining method for document understanding. It involves a new hierarchical model architecture and three pretraining tasks. It compares with previous SOTA methods on three tasks and achieves stronger performance on two of them.

The major contribution is the newly proposed multi-modal pretraining tasks and the architecture is a combinations of models from both domains.

**Limitations And Societal Impact:**

To the best of my knowledge, the authors have identified and discussed the potential social impact of this work.

**Main Review:**

**Originality**: Apart from some novelties of the pretraining objectives, the method reuses some of the existing architectures in a hierarchical way. However, it achieves significant improvements of two of the proposed tasks.

**Quality and Significance**: Overall, this is a good application method and might be useful for many practical applications.

**Clarity**: The writing needs to be improved. Many parts of the method are described verbally instead of formal definitions, making the paper hard to read. Some of the terms also need to be defined before using, such as RoI features, semantic regions.

**Time Spent Reviewing:**

2

---

> ### Author Response · Authors · 2021-08-10
> **Response to Reviewer Zk5e**
>
> We thank the reviewer for the positive feedback and suggestions.
>
> ### **Q: Writing needs to be improved and some of the terms also need to be defined before using.**
>
> We thank the reviewer for the suggestions on writing.
>
> **[1]** Region of Interest (RoI) features are computed by applying RoI Align (extracting the feature map for each bounding box) and RoI Pooling (producing the fixed-size features by applying max-pooling on the feature map extracted by RoI Align), where RoI Align and RoI Pooling [A] are the operations widely used in object detection tasks using convolutional neural networks.
>
> **[2]** The semantic regions considered by UniDoc are close to the object-level semantic elements in natural images. For documents, we represent the typical document layout elements such as paragraph, title, figure, and table as semantic regions. This is also well-aligned with human reading behaviors. When we humans read a paper, the title, paragraphs, and figures are the first elements we pay attention to. Independent words only have value when considering the context; that is the reason we select the semantic region as the inputs for UniDoc. We will define these terms before using them and further polish the writing of our paper.
>
> [A] Girshick, Ross. "Fast R-CNN." Proceedings of the IEEE international conference on computer vision. 2015.

---

### Official Review · Reviewer_5oFD · 2021-07-15

**Rating:** 4
**Confidence:** 2

**Summary:**

This paper proposes a new unified framework for pre-training image/text encoders for capturing both visual and textual features in documents. The main Transformer model is pre-trained with multimodal embeddings as input features (visual & textual features) through three pre-training tasks: Masked Sentence Modeling, Visual Contrastive Learning, and Vision-Language Alignment. The pre-trained model can be fine-tuned on downstream tasks on which the proposed UniDoc model outperforms baselines.

**Limitations And Societal Impact:**

I think the authors have made reasonable responses regarding the limitations and potential negative social impact of their work.

**Main Review:**

[Originality]

I feel that the methods/model architectures are largely putting together existing pieces of work from vision and language fields. In Sec 3.1, the proposed feature extraction, quantization, and gated cross-attention modules are all rather straightforward variations of existing work (i.e., the essential mechanisms are not new). Although they do make sense and are reasonable choices for a multimodal framework, I see little novelty here. In Sec 3.2, the three pre-training tasks also resemble well-known techniques, like masked language modeling in NLP and contrastive learning in vision. The alignment of vision and language is also not particularly surprising/insightful and provides little performance gain according to Table 2.

[Quality]

I do not find anything wrong with the proposed methods and models, since they are largely borrowed from well-known existing work. However, the overall technical contribution does not seem exciting/insightful to me. One thing that appears weird to me is the frozen sentence encoders initialized with the weights trained on NLI and STS datasets--if they have been fine-tuned on those tasks, they may capture too corpus-specific features to generalize to other domains (like the new pre-training documents). I don't think the computational constraints are valid excuses if the paper is proposing a new pre-training framework, which should be generic and not task-specific.

[Clarity]

The paper is overall well-structured and the methods are explained clearly.

[Significance]

The effectiveness of the proposed model is evaluated on four downstream tasks, by comparing with a set of baselines. The model outperforms the baselines with large margins. However, I am not sure how significant the results are to the NeurIPS community, because none of the baselines compared, as far as I am aware of, are from ML conferences. I would like to see comparisons with stronger baselines (e.g., there are many better PLMs than BERT from the language field), and ideally how the framework performs on pure language applications (e.g. the GLUE & SQuAD benchmarks) since a lot of applications may not have images and texts simultaneously. If the UniDoc model cannot be effectively used for pure language applications, I am not convinced that this can be called a "unified" framework.

[Summary]

Pros:
* The methods are clearly introduced with reasonable motivation.
* The connection with existing work is described in detail.
* The model has better empirical performance than a set of baselines on some specific applications.

Cons:
* The methods are not novel/insightful as they are largely borrowing from existing work.
* There are concerns about the generalization ability of the model since it uses dataset-specific text encoders.
* The significance of the empirical contribution made is not clear to me (the baselines are not from the ML community and are not strong enough; it's not clear whether the model works for pure language applications as a "unified" framework)

**Time Spent Reviewing:**

7

---

> ### Author Response · Authors · 2021-08-10
> **Response to Reviewer 5oFD**
>
>
> Thank the reviewer for the thoughtful comments.
>
> ### **Q: Concern for contribution and novelty.**
>
> UniDoc indeed adopts some existing techniques (transformer and contrastive learning). However, as a pretraining framework for document pretraining, it is new.
>
> Firstly, UniDoc is a multimodal end-to-end pretraining framework for document images.
> We would like to point out that UniDoc was designed for document image pretraining. The special characteristics of document images (spatial layout and multiple elements) distinguish document image pretraining from pretraining works in NLP and vision-language domains. In the NLP domain, the inputs are pure texts without spatial regions (bounding boxes). In the vision-language domain, the inputs are the detected objects (with bounding boxes) and caption texts. While for document images, the input elements (textual and visual) are spatially distributed. The techniques in UniDoc, i.e., semantic region inputs, hierarchical transformer encoder (sentence encoder+gated cross-attention, etc.) are specifically designed to handle various types of documents and document-related applications. This is one of the key ways in which our work is different from previous works (LayoutLM, SelfDoc, etc.)
>
> Secondly, the design of pretraining tasks for UniDoc is non-trivial. It is true that our pretraining tasks are related to the popular paradigm of learning by masking. However, we would like to highlight the differences: Firstly, our masked sentence learning is rather different from masked language modeling; instead of masking the input words, we mask the sentence embeddings and let the model reconstruct the masked embeddings. Secondly, applying the ideas of vector quantization and visual contrastive learning to document pretraining is new. To the best of our knowledge, our paper is the first successful attempt to introduce the quantization idea into multimodal pretraining for document images.
>
> Lastly, UniDoc can be applied to different kinds of document applications. For example, the hierarchical transformer encoder can easily be extended to multilingual (using pretrained sentence encoder in different languages) and multi-pages (region-based design can effectively handle long-range inputs) settings. Also, as shown in the experiments, the pretrained visual backbone can hugely boost the performance of document object detection. By adopting different visual backbones (ResNet, MobileNet, U-Net, etc.) during pretraining, we can apply UniDoc to various document image recognition tasks (OCR, Detection, Segmentation, etc.)
>
> ### **Q: Using frozen sentence encoder during pretraining and finetune the sentence encoder.**
>
> Nice comment. The reasons we freeze the sentence encoder are: 1) We adopt the sentence BERT as a pure-text understanding tool to process the OCR words, which maps sentences \& paragraphs to semantically meaningful sentence embeddings. We agree that the sentence encoder trained on NLI may bring some corpus-specific features to UniDoc. However, we have a cross-attention encoder that comes after the sentence encoder, which can learn the refined and more generalizable features from pretraining document images. 2) During pretraining, we mask the sentence embedding and predict the masked sentence embedding, while keeping the sentence encoder frozen ensures that the 'target' feature space is fixed and eases the pretraining process. Actually, we did try unfreezing the sentence encoder during pretraining, but the constantly changing sentence encoder caused instability in both the 'target' feature space and the pretraining process.
>
> Since different tasks require task-specific input embeddings to perform well, instead of finetuning the sentence encoder during pretraining, we explored unfreezing the sentence encoder during the finetuning stage to help the model learn task-specific sentence embeddings. Unsurprisingly, we see some performance improvements on several downstream applications, FUNSD (87.96$\rightarrow$87.93$\downarrow$), CORD (98.85$\rightarrow$98.94$\uparrow$), RVL-CDIP (93.96$\rightarrow$95.05$\uparrow$). However, this also increases the learnable network parameters and makes the training more challenging in terms of computational resources and training time.
>
> ### **Q: Stronger baselines and none of the baselines compared, as far as I am aware of, are from ML conferences.**
>
> Although document understanding has been studied for a long time, document image representation learning is a new topic. The lack of training data and annotations becomes a hindrance for researchers working in this area. Thanks to the recent advances in self-supervised learning, there are a few recently proposed methods on the topic (we have described them in the related works and made the experimental comparison with them in our paper, please see Table 3), and they are published as part of the proceedings of top-tier machine learning conferences in computer vision and NLP. Some examples of the published papers are LayoutLM (KDD 2020), LayoutLMv2 (ACL 2021), and SelfDoc (CVPR 2021).
>
> ### **Q: Pure language applications, I am not convinced that this can be called a "unified" framework.**
>
> First, we would like to point out that our main focus is on document image understanding tasks, and UniDoc was designed to handle document (image) applications, not pure-text/NLP tasks (e.g., GLUE & SQuAD benchmarks). There are huge differences and domain gaps between the traditional text-only document (1D) in NLP and the document images (2D) in our setting. Second, even though pure NLP applications are not the focus of this paper, we did experiment with a pure language-based UniDoc as shown in Table 2. UniDoc (Language-only, removing the visual branch and replacing the cross-attention with self-attention. Please see L292-L296.) with a hierarchical encoder can also be applied to pure-text/NLP applications.

---

> > ### Comment · Reviewer_5oFD · 2021-08-25
> > **Thank you for the response, but not fully convinced**
> >
> > I would like to thank the authors for their response. However, my major concerns about this paper have not been resolved:
> > * Novelty: There seems to be a consensus among all reviewers that the paper is largely putting together existing methods/techniques to form the new framework. Although I acknowledge that the joint modeling of textual content and images has some differences from separately modeling them using NLP/CV techniques, the methods used for the joint modeling still highly resemble well-known approaches (e.g., masked sentence learning and the application of vector quantization and visual contrastive learning are all slight variations of previous methods, instead of something fundamentally new). Therefore, the major contribution of this paper is on the application side, which should be suitable for a more application-oriented venue (like KDD, ACL, or CVPR, which is also why the baselines are from these conferences). For an ML-focused conference like NeurIPS, I expect to see more insightful/novel ML techniques introduced.
> > * Significance and paper claims: The paper is positioned as a "unified framework" (Note that the baseline multimodal pretraining frameworks do not make such claims)--given this claim, the model should be able to solve any task that can be done with pure language models (especially considering the authors put "BERT" as a keyword of the submission), though it is acceptable if its performance does not achieve the state-of-the-art compared to pure language models. Therefore, omitting experiments on well-established language benchmarks (like GLUE/SQuAD) and ignoring stronger PLM baselines than BERT (e.g. RoBERTa and ELECTRA) do not show good scholarship.
> >
> > Given the above consideration, I would like to maintain my original rating.

---

> > > ### Author Response · Authors · 2021-08-27
> > > **Thanks for your comments**
> > >
> > > **[1]** We respectfully disagree with the comments on putting together existing methods and the incompatibility of our paper with NeurIPS. Firstly, as mentioned in the paper and response, the framework and pretraining tasks (e.g., vector quantization)  proposed in this paper are new for representation learning in the document understanding domain. Secondly, unlike the vision/NLP/vision-language domains, our paper is designed for document image understanding, and this is a new and important topic; there are very few works push the frontiers in this direction. The "*Call For Papers*" of NeurIPS 2021 explicitly mentions the topics covered by the conference that include *deep learning and applications*, and we think our paper fits very well to such topics.
> > >
> > > **[2]** We also strongly disagree that we should include pure-NLP problems. As mentioned in our paper, we work on document images, which are totally different from the data in pure-NLP (1D) tasks. UniDoc is designed for document image understanding, and we deem UniDoc as a unified approach because it alone can serve and benefit many document image understanding tasks (form understanding, receipt understanding, document classification, and document detection, etc.), not because it can work across multiple different modalities (pure computer vision, pure NLP). We do mention UniDoc is not designed for pure NLP tasks, and thus it is not fair to compare document images with pure-text since document images (that contain both visual and textual information) are totally different from the pure texts in NLP domain.
> > >
> > > **[3]** We want to point out that it is unreasonable to experimentally compare UniDoc with RoBERTa and ELECTRA.  Firstly, the pretraining data and tasks of RoBERTa and ELECTRA are vastly different than those of UniDoc, given that they have different goals and solve different problems than UniDoc does. RoBERTa and ELECTRA are mainly designed for pure-text data. Specifically, RoBERTa enhances BERT with improved training methodology (dynamic masking, larger batch-training size, without NSP loss, etc.). ELECTRA comprises a BERT-based generator network and BERT-based discriminator (replaced token detection, etc.). Such pure-text pretraining tasks of RoBERTa and ELECTRA cannot be directly applied to document images. Secondly, UniDoc is fundamentally different from RoBERTa and ELECTRA, both in terms of data modality and the granularity level of input tokens. RoBERTa and ELECTRA are both variations of the traditional BERT, and they work on a single modality using word-level inputs, whereas UniDoc takes multimodal features at the document-region level as inputs and encodes them with gated cross-attention.

---

### Official Review · Reviewer_xpLb · 2021-07-19

**Rating:** 6
**Confidence:** 4

**Summary:**

This paper proposes a self-supervised pre-training framework for document understanding called UniDoc. UniDoc takes multimodal data (image feature and text feature from document) as input and uses gated cross attention for learning cross-modal correlation. The superiority of UniDoc has been empirically shown on several downstream tasks.

**Limitations And Societal Impact:**

I think interesting future research directions are described in the last section of the paper.

**Main Review:**

This work proposes a pre-training method for document understanding by designing a model architecture and training objectives that can effectively utilize visual features as well as textual features.
UniDoc achieves remarkable performance improvement in downstream tasks such as entity recognition, document detection, and reception understanding. Extensive experiments and ablation studies show that each component is beneficial.

While used components are effective, each of them is from other works. In other words, UniDoc is a combination of well-known techniques, having a lack of significant novelty.

There is no performance comparison with word-level pre-training. Since the entity recognition task is word-oriented as mentioned in the paper, it would be better to perform pre-training on word-level rather than sentence-level.
UniDoc does not improve the performance of document classification tasks over previous methods. Although the authors argue that it attributes to the difference of OCR engines, they should explain the magnitude of this difference. They say the difference between EasyOCR and Tesseract, but should report performance metric and qualitative analysis on the RVL-CDIP dataset to compare real Microsoft OCR and EasyOCR.

**Time Spent Reviewing:**

7 hours

---

> ### Author Response · Authors · 2021-08-10
> **Response to Reviewer xpLb**
>
> We thank the reviewer for the positive feedback and insightful comments.
>
> ### **Q: Lacking of significant novelty.**
>
> Although the techniques (i.e., transformer, contrastive learning, etc.) adopted in our paper already exist, learning contextual visual and textual information in a single unified framework is new. Moreover, it is non-trivial to design and train such a framework. Different from previous pretraining frameworks in the document, vision-language, and NLP domains, UniDoc has several key differences and contributions:
>
> **[1]** Although BERT-based pretraining has been explored for a long time, pretraining for document images is still far from mature due to its unique characteristics and challenges. Unlike the BERT-based models in vision-language and NLP, what distinguishes UniDoc from previous works is the unified multimodal pretraining framework. The design of the three pretraining tasks, as well as their effects on the downstream performance, are also non-trivial, especially the visual branch.
>
> **[2]** Unlike the vision-language domain, where there is well-defined visual information (object labels) used in pretraining, there is no clear and definite answer to "which kind of visual information is useful for document understanding?''. To the best of our knowledge, we are the first to bring the quantization idea into document pretraining. Also, to enforce that the multimodal information is effectively modeled and utilized for the regions in each document image, we specifically design the gated cross-attention and three pretraining tasks.
>
> **[3]** Lastly, the "Unified'' idea is new to the document domain and is vital for document understanding. UniDoc is trained in an end-to-end fashion. It extracts the words via OCR and encodes them with a publicly trained sentence encoder, making it possible to handle document images in different languages by replacing the sentence encoder in other languages. Also, the hierarchical design (sentence encoder + cross-attention document encoder) enables UniDoc to handle long documents, even for the multi-page ones.
>
> ### **Q: Word-level pretraining.**
>
> Nice comment. It is true that word-level pretraining has dominated most of previous pretraining works (document, NLP, and vision-language). Our response is as follows:
>
> **[1]** Why not include word-level pretraining?
> First, we think document pretraining should be treated differently than sequence-based pretraining in NLP, since the words in the document (2D) are arranged according to spatial layouts, while words in NLP corpora are sequential (1D). Considering the special characteristic of documents (complex layout, multi-pages) and the limited input length that BERT model is able to handle, it is not trivial to directly formulate the input at the word level. Base on the above, UniDoc introduces the ideas of modeling semantic regions and hierarchical encoding. Secondly, although we encode words with a sentence encoder, a large portion of extracted OCR regions have single words. We analyze the distribution of words (Easy OCR) per region for RVL-CDIP (a subset of IIT-CDIP) according to the categories as follows:
>
> |\# words/region | Letter | Form | Email | Handwritten | Advertisement | Scientific report | Scientific publication |Specification|
> |----|--------|----------|---------|---------|---------|----------|-----------|--------|
> |1 | 24\% |33\% |25\% |24\% |31\% |32\% |28\% |32\%|
> |2- 10 | 49\% |52\% |41\% |50\% |49\% |50\% |34\% |56\%|
> |>10 | 27\% |15\% |34\% |26\% |21\% |17\% |38\% |11\%|
> |**\# words/region** | **File folder** | **News article** | **Budget** | **Invoice** | **Presentation** | **Questionnaire** | **Resume** | **Memo** |
> |1 |50\% |27\% |41\% |38\% |24\% |31\%  |28\% |35\%|
> |2-10 |47\% |41\% |52\% |50\% |50\% |51\% |43\% |44\%|
> |>10 |3\% |32\% |8\% |12\% |26\% |18\% |30\% |21\%|
>
> It can be seen that despite the fact that we consider region-level input, for some semi-structured documents, single-words dominate the inputs; this somehow forces UniDoc to consider the word-level inputs. Different from traditional BERT work that predicts the masked word with MLM, UniDoc predicts the textual embedding of the masked input with MSM.
>
>
> **[2]** What if word-level pretraining is included? As suggested, we consider masked token modeling during pretraining and conduct the pretraining and finetuning experiments as follows: a) Since the number of words may be very long, we select the tokens by randomly applying sliding window (window size 128) across all OCR words. b) We formulate each word as a single-word sentence ([CLS] [Token] [SEP]). We randomly mask 15\% of those sampled words ([CLS] [MASK] [SEP]) and concatenate them along with the region-based inputs. c) During pretraining, we add a word prediction head on top of UniDoc, and predict the masked token with cross-entropy loss. d) Due to time limitations, we use 300k pretraining samples (the same as Table 2), and such a direct combination hurts the performance: FUNSD: 87.38 (UniDoc) vs. 83.76$\downarrow$ (UniDoc+MLM), CORD: 98.75 (UniDoc) vs. 98.63$\downarrow$ (UniDoc+MLM). Since we have word and RoI features for each token, the RoI features extracted by token bounding boxes might not be discriminative enough and negatively impact the model due to the tiny word bounding boxes. It may be worthwhile to consider a more extended region during the word-level RoI feature extraction. Also, it is beneficial to explore better pretraining tasks to make MLM work better, such as modeling the relationship between the words and regions (since any word always belongs to a region).
>
> ### **Q: Document classification and the importance of OCR.**
>
> Thanks for this comment. Instead of extracting the pretraining corpus, we only extract the RVL-CDIP with Google OCR and finetune our model on the OCR results. We initialize the model with pretrained UniDoc (Easy OCR) and finetune on Google OCR results, and the report the results with different OCR:  93.42 (Tesseract OCR) $\rightarrow$ 93.96 (Easy OCR) $\rightarrow$ 94.10$\uparrow$ (Google OCR). We further finetune the sentence BERT and achieve a performance boost: 93.96 (Easy OCR) $\rightarrow$ 95.05$\uparrow$(Google OCR). We will report the experiments (pretraining and finetuning) with Google OCR in the revised version.

---

### Decision · Program_Chairs · 2021-09-27

**Decision:**

Accept (Poster)

**Comment:**

This paper presents a new framework for pre-training for document understanding tasks like document and entity classification. The framework combines visual information from the input document image with textual information from OCR output using several self-supervised objectives. Results demonstrate improvements over state-of-the-art document encoding baselines on several downstream tasks.

The majority of reviewers are in favor of acceptance. Generally, reviewers praised the paper's approach as well-motivated and clearly described, and results as compelling. Several reviewers question the novelty of the ML contribution of this paper. I discount this criticism to some extent given that this is clearly a novel *application* of existing pre-training components, arranged into a new framework, demonstrating gains on important downstream tasks -- this seems squarely fair game for NeurIPS.

Other more minor criticisms include:

(1) Concerns about outdated baselines, e.g. why not use more recent MLMs as baselines like RoBERTa and ELECTRA? Rebuttal argues that these are not comparable because they focus on text representation alone. I don't completely buy this since BERT was used as a baseline.

(2) Some concerns about clarity (more formal mathematical descriptions would help) and ablations (there are many moving parts here, which model components are indispensable?).

(3) One reviewer was concerned that there were no comparisons with word-level pre-training -- author response points out that region-level often is word-level, but also include additional experiments with word-level MLM for comparison showing worse performance. Thus, I believe this concern was mostly addressed in rebuttal.

(4) Two reviewers were concerned that there was no improvement on document classification over the top baseline. This has been resolved in author response with new experiments demonstrating the effect of underlying OCR system.

Overall, I agree with the majority of reviewers and recommend acceptance. The issues mentioned by reviewers (and summarized above) absolutely need to be addressed in revision -- e.g. further ablations, improvement to mathematical clarity, comparisons with more recent MLMs, and the inclusion of all additional experiments provided in rebuttal. However, in this case, I do not believe the revisions are so substantial as to warrant another round of review.